



# Soil Moisture Sensor Network Design for Hydrological Applications

Lu Zhuo[1,2], Qiang Dai[2,3*], Binru Zhao[4], Dawei Han[5]

[1]Department of Civil and Structural Engineering, University of Sheffield, Sheffield, UK

[2]Key Laboratory of VGE of Ministry of Education, Nanjing Normal University, Nanjing, China

[3]Jiangsu Center for Collaborative Innovation in Geographical Information Resource Development and Application, Nanjing, China

[4]College of Water Conservancy and Hydropower Engineering, Hohai University, Nanjing, China

[5]WEMRC, Department of Civil Engineering, University of Bristol, Bristol, UK

[*]Correspondence: qd_gis@163.com

**Abstract**

Soil moisture plays an important role in the partitioning of rainfall into evapotranspiration, infiltration and runoff, hence a vital state variable in the hydrological modelling. However, due to the heterogeneity of soil moisture in space most existing in-situ observation networks rarely provide sufficient coverage to capture the catchment-scale soil moisture variations. Clearly, there is a need to develop a systematic approach for soil moisture network design, so that with the minimal number of sensors the catchment spatial soil moisture information could be captured accurately. In this study, a simple and low-data requirement method is proposed. It is based on the Principal Component Analysis (PCA) and Elbow curve for the determination of the optimal number of soil moisture sensors; and *K*-means Cluster Analysis (CA) and a selection of statistical criteria for the identification of the sensor placements. Furthermore, the long-term (10-year) soil moisture datasets estimated through the advanced Weather Research and Forecasting (WRF) model are used as the network design inputs. In the case of the Emilia Romagna catchment, the results show the proposed network is very efficient in estimating the catchment-scale soil moisture (i.e., with *NSE* and *r* at 0.995 and 0.999, respectively for the areal mean estimation; and 0.973 and 0.990, respectively for the areal standard deviation estimation). To retain 90% variance, a total of 50 sensors in a 22,124 km$^2$ catchment is needed,



which in comparison with the original number of WRF grids (828 grids), the designed network
requires significantly fewer sensors. However, refinements and investigations are needed to
further improve the design scheme which are also discussed in the paper.
**Keywords:** Soil moisture network design, Principal Component Analysis (PCA), *K*-means
Cluster Analysis (CA), Weather Research and Forecasting (WRF) Model, Optimising,
Numerical Weather Prediction (NWP) model.
**1.  Introduction**
Soil moisture is at the heart of the Earth system and it plays an important role in the exchanges
of water and energy at the land surface (Dorigo et al., 2017;Robock et al., 2000;Crow et al., 2018).
In hydrology, soil moisture is the key component for the partitioning of rainfall into
evapotranspiration, infiltration and runoff (Vereecken et al., 2008;Brocca et al., 2017;Rajib et al.,
2016;Fuamba et al., 2019). In particular, the antecedent soil moisture condition of a catchment is
among one of the most important factors for flood triggering (Uber et al., 2018;Zhuo and Han,
2017). For hydrological modelling, soil moisture is a vital state variable. Especially, during
real-time flood forecasting, the accurate updating of the soil moisture state variable is a critical
step to reduce the accumulation of model errors (i.e., time drift problem) (Lopez et al.,
2016;Laiolo et al., 2016;Zwieback et al., 2019). Therefore, the intensive monitoring of catchment-
scale soil moisture content would benefit a number of hydrological applications.
In-situ soil moisture sensors (e.g., capacitance probe, and Time Domain Reflectometry), as one
of the oldest and most common methods used around the world, can provide point-based soil
moisture measurements with relatively high accuracy in comparison with the modelling and
the remotely sensed approaches (Albergel et al., 2012). Therefore, they are a crucial source of
information for the hydrological research (Western et al., 2004;Brocca et al., 2017). However, due
to the heterogeneity of soil moisture in large space and the economic considerations, most





existing in-situ networks rarely provide sufficient coverage to capture the catchment soil
moisture variations (Chaney et al., 2015). In particular, in a number of cases, soil moisture
sensors are mainly installed close to the residential plain areas (e.g., due to easy accessibility
and maintenance reasons), and there is a lack of sensors installed in the complex topographic
areas where they are really the most needed (Zhuo et al., 2019b). Therefore, there is a need to
develop a systematic approach for the soil moisture network design, so that with the minimal
number of sensors the catchment-scale soil moisture information could be captured accurately.
However, to our knowledge, there is a lack of existing literature covering such a research area
particularly for the hydrological applications (Chaney et al., 2015), albeit numerous studies have
been carried out on the rain gauge network design by the community (Dai et al., 2017;Adhikary
et al., 2015;Pardo-Igúzquiza, 1998;Chen et al., 2008;Bayat et al., 2019).
Therefore, to address the aforementioned research gap, the aim of this paper is to propose a
pioneer soil moisture network design scheme for catchment-scale studies, based on a
combination of statistical approaches. In particular, the Principal Component Analysis (PCA)
and Elbow curve are adopted to determine the optimal number of soil moisture sensors within
a catchment, and *K*-means Cluster Analysis (CA) and a selection of statistical criteria are used
for the identification of the soil moisture sensor placements. Although the methodologies
themselves are not new, it is the first time they are applied for the soil moisture network design.
Furthermore, long-term (10-year) soil moisture datasets estimated through the advanced
Numerical Weather Prediction (NWP) Weather Research and Forecasting (WRF) model
(Skamarock et al., 2008)  are used as the design inputs. WRF model has been applied in a wide
range of applications with good performances (Srivastava et al., 2015;Zaitchik et al., 2013;Zhuo et
al., 2019a;Stéfanon et al., 2014). Although WRF estimated soil moisture cannot represent the
ground truth, they are ideal datasets to provide catchment characteristics, such as land cover,
soil properties, topographies, which are the main drivers of local soil moisture heterogeneity



(Friesen et al., 2008). Therefore, such globally available datasets together with the proposed
statistical approaches would provide useful insights for the soil moisture network design
research (i.e., to minimise the redundancy of information, and improve accuracy), in particular,
for those currently ungauged catchments. In this study, the proposed method is implemented
in the Emilia Romagna region, northern Italy as a case study due to its high-exposure of flood
events.
The paper is organised as: the study area is introduced in Section 2; soil moisture network
design methodologies are described in Section 3; the results are presented in Section 4; and
discussions and conclusions are included in Section 5.
**2.  Study Area**
In this study, the Emilia Romagna region (latitude 43°50ʹN–45°00ʹN; longitude 9°20ʹE–12°40ʹE)
is selected for the case study which is in Northern Italy (Figure 1). The region's total coverage
is approximately 22,124 km$^2$. It is surrounded by the Apennines to the south and the Adriatic
Sea to the east, with over half of the area as a plain agricultural zone (12,000 km$^2$). The climate
condition is highly varied in the region which is largely influenced by the mountains and the
sea, with subcontinental in the Po Plain and hilly areas, and cool temperate in the mountain
range (Nistor, 2016). It has distinct wet and dry seasons (i.e. dry season between May and
October, and wet season between November and April) (Zhuo et al., 2019b). Based on the ESA
CCI land cover map (Bontemps et al., 2013), the region is mainly covered by Herbaceous (37%),
followed by Tree (22%), and Cropland (21%). The majority of the area is on the quaternary
alluvial deposits, which are characterised by a high degree of heterogeneity (Pistocchi et al.,
2015). The annual temperature ranges from 8.2 to 19.3°C; and the annual mean precipitation is
between 520 and 820 mm (Pistocchi et al., 2015).



For the soil moisture network in the region, currently, there is a total of 19 soil moisture sensors
installed (all located in the plain area); however only one of them can provide long-term
continuous soil moisture monitoring datasets. The network is managed by the Regional Agency
for Environmental Protection Emilia Romagna Region. Through further investigations, it has
been found, a number of the sensors have actually never provided proper soil moisture
measurements since the installation. For such a highly heterogeneous catchment, only one soil
moisture sensor at the plain area is clearly not sufficient for any catchment-scale applications.
Therefore, it is hoped the proposed soil moisture network design scheme could provide some
useful guidance to the local authority on an improved network in the future (i.e., a minimum
number of sensors for reduced installation and maintenance cost, but at the right locations).
**3.  Methodologies**
**3.1 WRF Model**
The WRF model is a next-generation, non-hydrostatic mesoscale NWP system designed for
both atmospheric research and operational forecasting applications (Skamarock et al., 2005). The
model is capable of modelling a wide range of meteorological applications varying from tens
of metres to thousands of kilometres (NCAR, 2018). Apart from the WRF's aforementioned
advantage on including the catchment characteristics for the soil moisture estimations, it also
has other merits that make it an ideal tool for providing the distributed soil moisture information
for the network design. For instance, WRF model's spatial and temporal resolutions can be
changed depending on the input datasets to fit various application requirements, and a number
of globally-available data products can be selected to provide the necessary boundary and
initial conditions for running the model. Therefore, WRF is able to provide valuable
information for this study. Here WRF version 3.8 with the ARW dynamic core is used.
**3.1.1  Model Parameterization**



Apart from the atmospheric forcing, parameterization is also required to drive the WRF model.
In particular, the microphysics scheme is important in simulating accurate rainfall information
which in turn is significant for estimating the accurate soil moisture fluctuations. WRF V3.8
supports 23 microphysics options ranging from simple to more sophisticated mixed-phase
physical options. In this study, the WRF Single-Moment 6-class scheme is adopted which
considers ice, snow and graupel processes and is suitable for high-resolution applications (Zaidi
and Gisen, 2018). The physical options used in the WRF setup are Dudhia shortwave radiation
(Dudhia, 1989) and Rapid Radiative Transfer Model (RRTM) longwave radiation (Mlawer et
al., 1997). Cumulus parameterization is based on the Kain-Fritsch scheme (Kain, 2004b) which
is capable of representing sub-grid scale features of the updraft and rain processes, and such a
feature is useful for real-time modelling (Gilliland and Rowe, 2007). The surface layer
parameterization is based on the Revised fifth-generation Pennsylvania State University–
National Center for Atmospheric Research Mesoscale Model (MM5) Monin-Obukhov scheme
(Jiménez et al., 2012a). The planetary boundary layer is calculated based on the Yonsei
University scheme (Hong et al., 2006a). In WRF, its land surface model plays a vital role in
the integration of information generated through the surface layer scheme, the radiative forcing
from the radiation scheme, the precipitation forcing from the microphysics and convective
schemes, and the land surface conditions to simulate the water and energy fluxes (Ek et al.,
2003). In this study, the Noah Multiparameterization (Noah-MP) is chosen, because it has
shown more accurate soil moisture estimation performance than the other two main schemes
(Noah and CLM4) in other studies (Cai et al., 2014;Zhuo et al., 2019a). Table 1 shows the selected
WRF parameterization schemes. The static inputs (i.e., land use and soil texture) are chosen in
the WRF pre-processing package. Here, the land use categorisation is interpolated from the
MODIS 21-category data classified by the International Geosphere Biosphere Programme



(IGBP). The soil texture data are based on the Food and Agriculture Organization of the United
Nations Global 5-minutes soil database.
**3.1.2    Model Setup**
The WRF model is centred over the Emilia Romagna Region, and integrates three nested
domains (D1, D2, D3), with the horizontal spacing of 45 km x 45 km (outer domain, D1), 15
km x 15 km (inner domain, D2), and 5 km x 5 km (innermost domain, D3). In this study, the
innermost domain D3 is used (88 x 52 grids (west-east and south-north, respectively)), with a
two-way nesting scheme considered letting the information from the child domain to be fed
back to the parent domain. To drive the WRF model, the European Centre for Medium-Range
Weather Forecasts (ECMWF) reanalysis (ERA-Interim) is adopted to provide the study
region's boundary and initial conditions. ERA-Interim is a global atmospheric reanalysis that
is available from 1979 to 2019 (ERA-5 as a recent update to ERA-Interim may also be used).
The spatial resolution of the datasets is approximately 80 km on 60 levels in the vertical from
the surface up to 0.1 hPa. It contains 6-hourly gridded estimates of three-dimensional
meteorological variables, and 3-hourly estimates of a large number of surface parameters and
other two-dimensional fields. Please see (Berrisford et al., 2011) for a detailed documentation of
the ERA-Interim.
After the initialization, the model needs to be spun-up to derive a physical valid state (e.g.,
equilibrium state) (Cai et al., 2014;Cai, 2015).  In this study, WRF is spun-up by running through
the whole year of 2005. After the spin-up, the WRF model is run in daily timestep from January
1, 2006, to December 31, 2015, using the ERA-Interim datasets. The modelled WRF grids
within the Emilia Romagna catchment (total of 828 grids) are shown in Figure 2 as black dots,
with the elevation map also illustrated in the background.
**3.2 Soil Moisture Network Design**

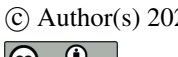



For the soil moisture network design, two main problems need to be tackled. First is how many
soil moisture sensors are needed within a catchment, and the second is where are the best
locations to place them. To solve the first problem, the PCA is used to obtain the optimal
number of soil moisture sensors through a threshold analysis. And for the second problem, the
$K$-means CA is adopted to determine the locations for the sensor placements.
**3.2.1. Principal Component Analysis (PCA)**
When soil moisture data are collected from $p$ soil moisture sensors, these data are often
correlated. This correlation reflects the complexity of the catchment and indicates that some of
the information collected from one sensor is also contained in the remaining $p$-1 sensors
(Gangopadhyay et al., 2001). The role of the PCA is to examine the redundancy of the WRF soil
moisture network, and more importantly to highlight the grids that provide the most significant
contribution to the principal components (Dai et al., 2017). The optimal number of sensors is
dependent on the amount of original variance the network should retain. PCA is a statistical
procedure for multivariance feature extraction. It adopts an orthogonal transformation to
convert a set of possibly correlated observations into a set of linearly uncorrelated variables
called principal components. This transformation is defined in such a way that the first principal
component has the largest possible variance, and each succeeding component in order has the
highest variance possible under the constraint that it is orthogonal to the preceding components
(Wold et al., 1987).
In this study, we have $p$ WRF soil moisture grids with $N$ observations (the time series of the
data, i.e., 10-year daily datasets). The covariance matrix $p$ x $p$ can be calculated which is
denoted as $X$, and the eigenvectors and the eigenvalues of the matrix can also be determined,
correspondingly. Since the eigenvectors of the $X$ are orthogonal, the $p$ eigenvectors are used to
construct the principal components, which can be represented as:





199          $\text{eigenvector} = (eig_1 \ eig_2 \ eig_3 \ \dots \ eig_p)$           (1)

with such a relationship, the original datasets can be transformed in terms of eigenvectors into
a new dataset $Z$. $Z$ is shown as the following:

202         $Z_i = X_1 eig_{i,1} + X_2 eig_{i,2} + \dots + X_p eig_{i,p} \ , \quad i = 1, \dots, p$       (2)

where $Z_i$ is the new dataset, $X_i$ is the original dataset. The variance of each of the component is
the eigenvalue. The eigenvector with the highest eigenvalue is the principal component of the
dataset. The examination of the network redundancy is implemented based on the desired rate
of variance contribution, and the number of principal components can thus be calculated
correspondingly. In other words, the appropriate number of soil moisture sensors are dependent
on the amount of original variance the network would like to retain. If for a specific desired
variance, the determined number of principal components ($k$) is significantly less than the total
number of the WRF soil moisture grids ($p$), then it can be concluded that the network is heavily
redundant, and even by removing a large number of grids, the remaining can still provide
sufficient soil moisture information for the entire catchment; and vice versa. In this paper, the
variance contribution rate of 70%~99% is tested. Generally, the required number of grids
increases when the variance contribution rate increases. However, the growth rate is not
constant that normally changes significantly at a critical point (threshold), which is used in this
study as the desired rate for the soil moisture network design.
**3.2.2. *K*-means Cluster Analysis (CA)**
After deciding the optimal number of soil moisture sensors from the PCA step, CA is then
applied to find the best locations for the sensors. CA is a multivariate method which aims to
classify a sample of objects into a number of groups so that similar objects are placed in the
same group (Everitt et al., 2001). The advantage of adopting the CA method for the network
design is that there is no prior knowledge required about which objects belong to which clusters.

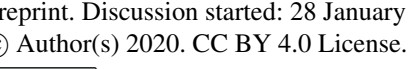



Because the optimal number of clusters ($k$) has already been determined by the PCA, $k$-means
clustering method is utilised in this study to divide the original $p$ datasets into $k$ clusters. $k$-
means approach is a typical distance-based clustering method which uses the distance as the
indicator for similarity among objects (i.e., the smaller the distance, the higher the similarity
of two objects) (Kodinariya and Makwana, 2013). In this study, the Euclidean distance is adopted
as the distance measurement. It is a simple and widely used way of calculating the distances
between objects in a multidimensional space (Danielsson, 1980). The centroid of each cluster is
the point which the sum of Euclidean distances from all objects in that cluster is minimized. It
is an iterative approach repeated for all of the clusters. Since an initial set of cluster centres is
needed to be given for the CA to start, the resultant performance will be sensitive to the initial
setting. In order to obtain an efficient performance, the WRF grids are ordered by their long-
term mean soil moisture and the initial cluster centres are selected evenly from the new
sequence (based on the number of $k$ from the PCA). After which, the WRF grids are attributed
to the closest cluster accordingly.

Within each of the optimised clusters, we propose two ways to find the most suitable grid for
the sensor placement. One way is by finding the grid which gives the median averaged soil
moisture in each of the cluster (denoted as CA-Med), and another is through identifying the
maximum averaged soil moisture in each of the cluster (denoted as CA-Max) (Dai et al., 2017).
As a result, for each cluster, there is one optimal grid, and grouped with the other optimal grids
found in other clusters, the ideal placements for the soil moisture sensors are identified. The
group of the selected grids is considered to be the optimal combination of locations that can
provide the desired variance of the original WRF soil moisture measurements over the whole
catchment.
**3.3 Network Evaluation**





Since there is no existing optimal in-situ soil moisture network that can be used as a reference
for the evaluation, it is challenging to assess the designed network performance based on a
comparison study. However, the designed network should be efficient enough to represent the
maximum amount of information with the minimum number of sensors within a catchment. In
other words, the designed network should retain the main catchment-scale soil moisture
information of the original WRF network, which is particularly important for the hydrological
modelling. To assess the network in such an aspect, the soil moisture information contained by
the designed and the original network are compared. Two statistical indicators are used for the
purpose, namely the Pearson correlation coefficient and the Nash–Sutcliffe coefficient.
The Pearson correlation coefficient ($r$) is a statistical measure of the linear correlation between
two sets of datasets, which in this study can estimate the systematic deviation between the
designed ($R_d$) and the original ($R_o$) catchment-scale soil moisture variations, and it is calculated
by the following equation:
$$r_{R_o,R_d} = \frac{E[R_d R_O] - E[R_d]E[R_O]}{\sqrt{(E[R_d{}^2] - E[R_d]^2) \times (E[R_O{}^2] - E[R_O]^2)}}$$
(3)

where $E$ is the mean value of the corresponding vector. In this study, the optimal performance
is achieved when $r_{R_o,R_d}$ equals to 1
Nash-Sutcliffe Efficiency (*NSE*) (Nash and Sutcliffe, 1970) is used widely in hydrology to
evaluate the prediction accuracy in hydrological modelling, which can be obtained by:
$$NSE = 1 - \frac{\sum \left(R_o^t - R_d^t\right)^2}{\sum \left(R_o^t - E[R_o]\right)^2}$$
(4)

where $t$ is the time-step of the dataset. The *NSE* ranges [1,-∞). The closer *NSE* is to 1, the more
accurate the designed network is.
**4. Results**



### 4.1. Soil Moisture Network Redundancy Analysis

Within the study area of 22,124 km$^2$, there is a total number of 828 WRF soil moisture grids. With such a dense network, there should exist information redundancy. To explore this, a cross-correlation ($r$) matrix for all of the grids over the whole study period is plotted in Figure 3. It can be seen that the majority part of the map is in blue-tone, which means most of the grids (85%) are correlated ($r > 0.5$) with the others (as shown in Table 2). In addition, over half of the grids (52%) have high correlation ($r>0.8$) with the rest of the grids; and even 15% of the grids can achieve very high correlation ($r>0.9$). However, it is clear from the map some grids (e.g., grid number 396-398, 523-529) are more heterogeneous than the others (red-tone, with low correlation <0.3 observed), which means more soil moisture sensors might need to be installed in those locations. The catchment map with the indicated WRF grid numbers is presented in Figure 4a). A further exploration of cross-correlation performance using box plots is shown in Figure 4b). The locations of the selected grids (as in Figure 4b) are marked in Figure 4a) with red circles. It can be seen the nine grids are distributed evenly within the catchment in order to represent a spectrum of catchment features (e.g., different land covers, elevations, soil types etc.). From the box plot, it can be seen for a specific grid, the cross-correlation can range from as low as below 0.1 to as high as almost 1. The large range is particularly obvious for Grid 500, which is located at the plain zone near the east boundary of the catchment and is close to the Valli di Comacchio lagoon. The closeness to the waterbody could mean its soil moisture is dominated more by the waterbody than by the local weather conditions, in comparison with grids located further away. For Grid 100, its correlation with the rest of the grids in the catchment is relatively low, with 75% percentile of the cross-correlations less than 0.6. The potential reason could be because it is located in the southern mountainous zone, with high-density of tree coverage and complex topographic conditions, its soil moisture is more heterogeneous than the other grids. A similar condition is observed for





Grid 1 which is also located in a hilly zone in the southern boundary of the catchment (i.e.,
lower correlation as shown in the boxplots). Such a phenomenon is not unexpected and could
mean more sensors are needed in those complex zones for better soil moisture monitoring
purpose. However, for Grids like 300, and 600 (and the surrounding areas), since the majority
of their correlations are high and they are located in plain areas with no water boundary nearby,
they could be arranged with a smaller number of soil moisture sensors.
**4.2. Soil Moisture Sensor Number**
In summary, through the cross-correlation exploration, many parts of the WRF soil moisture
network are significantly redundant, whilst for some parts, a denser network is indeed needed.
To systematically investigate the redundancy degree of the network, the PCA approach is
applied. Figure 5a) shows the PCA results to provide useful guidance on the acceptable loss of
information.  It is clear to see the first principal component carries close to 80% of the total
variance, with the second component bringing this to nearly 90%. This result again indicates
the high redundancy exists in the network, and just one component can contain almost 80% of
the total soil moisture information. To better understand the relationship between the principal
component numbers, the variance contribution rate, as well as the corresponding required grids
number, a set of variance contribution rates from 70% to 97.5% is used as the representatives.
The required number of components and the grids are listed accordingly in Table 3. It can be
seen only one component with 6 grids is sufficient to retain 70% of the soil moisture
information. Even when the variance is set at 80%, only two components are needed to meet
the requirement, and the corresponding number of soil moisture girds is 11 (1.3% percent of
the total grids). To satisfy 90% variance, three components are needed, and although the total
number of grids is increased to 50, it is still significantly less than the WRF's full inputs. The
detailed numbers further indicate the relatively high level of redundancy in the WRF's original
soil moisture network.

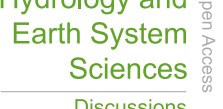

The trend can also be observed through the Elbow curve which is illustrated in Figure 5b). It
presents the relationship between the variance and the number of grids. It can be seen to meet
the increment of variance, the required number of grids also increases. But the growth rate is
the most significant when the variance is smaller than 70% and then slows down gradually
after that. When the variance meets 95%, the rate is further weakened. Based on the curve, it
is suggested the desired variance (i.e., trade-off point) between 80% and 95%. The required
number of soil moisture grids for 80%, 85%, 90%, and 95% is 11, 21, 50, and 184 respectively.
It is clear, in order to achieve the 95% variance, a significantly greater number of additional
grids are required, that is 268% more than for the 90% variance case. Therefore, for further
improvement of variance from 90% to 95%, the economic cost for the additional number of
sensors might not be as valuable as for the 85% to 90% case (138% additional sensors are
required for the enhancement).
**4.3. Soil Moisture Sensor Location Design**
Once the degree of redundancy for the full WRF soil moisture network is established, the next
step is to determine the optimal locations for sensor placements. Because the components from
the PCA do not directly represent the physical WRF grids, cluster analysis is thus carried out
to identify the specific grid locations. Here, CA-Max and CA-Med are used. The designed
networks for CA-Max and CA-Med are illustrated in Figure 6 and 7, respectively. The
indicated locations in the figures provide guidance on the preferential areas for the soil moisture
sensor placements. Each of the methods gives a different set of sensor locations, for instance,
the selected optimal soil moisture grids from the CA-Max method tend to be located at the
catchment boundary, and the situation is particularly obvious for the low variance cases (i.e.,
70% - 80%). For example, when the variance is set at 70%, the selected optimal locations from
the CA-Max is mostly distributed near the catchment's southern boundary, while from the CA-
Med, it is more homogeneously distributed (i.e., one at the southern boundary, one at the north,



two at the north-western part, and two at the north-eastern part). When the variance is increased,
for instance at 90%, the difference between the two CA methods becomes less distinctive.
Despite this, it can still be seen for the CA-Max, there is less coverage of sensors at the western
and the eastern parts of the catchment, with most of the sensors located at the mid-region.
However, for the same variance, the sensor distribution from the CA-Med looks more evenly
distributed visually. Nevertheless, when the variance reaches as high as 97.5%, the difference
from the two methods becomes rather small, as 367 sensors are located covering most parts of
the catchment in both cases.
**4.4. Soil Moisture Network Evaluation**
The evaluation of the designed network is challenging, as there are no standard assessment
criteria available to guide on what kind of network is the most appropriate for a given study
area. In essence, the designed network should be efficient, which means the network should
contain the maximum amount of information with a minimal number of sensors. In this study
since we focus on the soil moisture's hydrological applications (catchment-scale), to evaluate
the efficiency of the proposed schemes, the catchment-scale soil moisture data derived by the
designed networks are compared with the WRF's full inputs (828 grids). Both the areal spatial
mean and standard deviation are calculated. The Pearson correlation coefficient and the Nash–
Sutcliffe coefficient are used to quantify the relationships between the two soil moisture
datasets. The results for both the CA-Med and the CA-Max are compared in Figure 8. Based
on the areal mean soil moisture (Figure 8 a) and c)), it is clear to see the CA-Med outperforms
the CA-Max for the majority of the variance cases (both *NSE* and *r*), except for the *NSE* results
when the variance is over 90%. Moreover, for the *NSE* results, a decline of the performance
can be observed clearly after it passes the 90% variance point, which illustrates that an
increment of sensor number does not necessarily mean a arise of the performance. For the
standard deviation, the disparity between the two methods is smaller. When the variance is





below 80%, the growth trend for the CA-Med case is not clear, as it firstly drops at the 75%
point and then climbs up again when the variance increases. Whereas for the CA-Max case,
there is a clear upward trend. Similar to Figure 8 a), it is interesting to see for the areal standard
deviation in Figure 8 b) and d), the *NSE* and *r* also start to drop after reaching around 90%,
which again indicates the increment of sensor number does not positively link to the
improvement of network performance (here in the aspect of spatial variation). The evaluation
results are summarised in Table 4 for numerical comparison. Since CA-Med surpasses CA-
Max for most of the cases, it is chosen for the network design. In the aspect of the desired
variance, because as discussed earlier, when the variance climbs over 90%, the performance
instead drops. Therefore 90% variance is suitable to be used for the network design in this case.
The time series plots of the areal soil moisture mean and standard deviation are shown in Figure
9. Generally, the designed network can estimate the catchment's mean soil moisture very well,
as it follows the variation of the WRF's full input dataset closely ($NSE = 0.995$ and $r = 0.999$).
For the standard deviation, the general trend from both datasets shows a higher spatial variation
of soil moisture over the dry season and lower variation during the wet season. The spatial
variation is averaged around 0.04 $m^3/m^3$ throughout the whole study period. However, there
are some disparities between the two datasets, in particular, during the wet season (bottom
peaks in the STD plot), the designed network at several occasions overestimates the spatial soil
moisture variation, and during the dry season (top peaks in the STD plot), it underestimates
instead. Nevertheless, the differences are small and the correlation between the two datasets is
high, with $NSE = 0.973$ and $r = 0.990$ obtained.  In conclusion, the designed network can
maintain the dominated information of the WRF's full-grid input well.
The sensor displacements for the designed and the existing (in-situ) networks are illustrated in
Figure 10. In comparison with the distribution of the proposed network, the existing network
is clearly biased, with all of the sensors located in the mid-plain zone only. Such distribution





(i.e., no sensors located at the southern mountainous (highly-vegetated) region) can have
adverse impacts on the accuracy of the areal mean soil moisture estimation. Scatterplots of the
areal mean soil moisture calculated from the designed and the existing networks are also
presented in Figure 11. The performance difference between the two networks is clear to
observe. For the proposed network, the points are located close to the identical line, whereas
for the existing network, due to the inappropriate sensor distributions over the catchment, the
points are more dispersive ($NSE$ = 0.889). The performance of the existing network in
comparison with the proposed networks indicates that it cannot retain even 70% of the variance
(as compared with the $NSE$ results in Table 4), as the $NSE$ for the 70% CA-Med can achieve
0.949. For the existing network, without putting sensors in the highly vegetated region, the
network clearly underestimates soil moisture variations during the dry season (i.e., for the cases
when the soil moisture is less than 0.25 $m^3/m^3$)

**5. Discussions and conclusions**

With the low-cost soil moisture sensors becoming more and more available and modern
communication technology (i.e., Internet of Things), it is expected more in-situ soil moisture
sensors will be installed in the future. However, unlike the rich literature in the rain gauge
network design field, there is a research gap in soil moisture network design for catchment-
scale applications. As a result, research is urgently needed to fill this important knowledge gap.
As one of the pioneering studies in this field, a low-data requirement method is proposed in
this study for the in-situ soil moisture network design. Through a series of evaluations of the
developed network, it can be concluded that the method can provide efficient catchment-scale
soil moisture estimations (i.e., high accuracy of the areal mean and standard deviation soil
moisture estimations). To retain 90% variance, a total of 50 sensors in a 22,124 $km^2$ catchment
is needed. In comparison with the original number of WRF's grids (828 grids), the proposed
network requires significantly smaller number of sensors. Furthermore, in comparison with the



existing soil moisture network in the Emilia Romagna region, the proposed network has sensors
more evenly distributed, covering most representative parts of the catchment (e.g., both plain
and mountainous regions), and can obtain more accurate catchment-scale soil moisture
estimation. However, there are several points need to be discussed as follows.
The first point is about the uncertainty of the WRF's soil moisture estimations, which could
influence the accuracy of the network design. It is acknowledged that the reliability of the
designed network is influenced by the performance of the WRF model. To evaluate the WRF
results and test whether the proposed network can produce the catchment-scale soil moisture
well, a long-term densely covered soil moisture network will be required. Setting up such a
network is challenging and difficult to realise due to the high installation and maintenance cost.
In this study, a long-term WRF soil moisture estimation with 1-year spin-up time is used which
could to some extent produce a more stable result. But since "all models are wrong" (by George
E. P. Box), an uncertainty model (Zhuo et al., 2016) could be proposed to be integrated with the
network design scheme. For example, we can generate a large number of probable "true soil
moisture" datasets based on the proposed uncertainty model so that a set of possible soil
moisture networks can be produced. As a result, the designed network will be expressed in a
probabilistic form instead of a determinate form. In addition, a decision-making scheme
considering different conditions (e.g., accessibility, installation and maintenance cost) under
the uncertainty can be developed to select the most suitable soil moisture network. The
uncertainty influence of the WRF soil moisture on the network design will be investigated in
future studies.
Second, the case study is based on the daily soil moisture inputs for the hydrological
applications. With different research needs (meteorology, climatology, hydrology, water
resources, geology, etc.), various temporal-scale of soil moisture data might be required, for
example, climate change study requires soil moisture data in decades or hundreds of years





which often needs annual-scale measurements; drought assessment requires monthly to
seasonal datasets; while for hydrometeorological prediction applications, hourly datasets might
be needed. For the network design, the input data's temporal scale (daily, weekly, monthly,
yearly) can influence the final network design, therefore it is worth investigating in future
studies about the temporal-scale effect on the network design.
Third, for a complex catchment like Emilia Romagna, other uncertainty sources apart from the
WRF model can also affect the performance of the designed network; for instance, the study
area has varied climate conditions (a mixture of subcontinental and cool temperate) and distinct
seasonal changes (wet/dry seasons). Therefore separating/combining networks under different
catchment conditions could result in an improved soil moisture network design. Furthermore,
the poor accessibility to sensors is another challenging point that can hamper the performance
of the designed network in real life, for instance, even an in-situ network follows tightly
through a systematic design scheme, without proper maintenance due to the accessibility issue,
the quality of the retrieved data can be highly affected. Therefore, the accessibility factor
should also be considered for the network design (e.g., can be considered during the CA for
the sensor placements).
Since the forcing data for the WRF model is globally covered, the proposed scheme can largely
benefit ungauged catchments. On the other hand, in places where dense soil moisture networks
are already available, the proposed scheme could also help in minimizing the cost by reducing
the number of sensors. Another advantage of the method is that the number of soil moisture
sensors can be changed based on different variances to meet various requirements. Through
selecting different variance levels, the redundancy of the WRF's full-input network can be
assessed, and the corresponding optimal sensor number can be determined. However, the
proposed scheme is still in its infancy with a lot of refinements and further explorations needed,



therefore it is hoped this paper will stimulate more studies by the community in tackling the
soil moisture network design problem.
**Acknowledgement**
This research is supported by the National Natural Science Foundation of China (NSFC, grant
no. 41871299), and Resilient Economy and Society by Integrated SysTems modelling
(RESIST), Newton Fund via Natural Environment Research Council (NERC) and Economic
and Social Research Council (ESRC) (NE/N012143/1).

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



**Table 1.** WRF parameterizations used in this study.

|  | Settings/ Parameterizations | References |
| --- | --- | --- |
| Map projection | Lambert | |
| Central point of domain | Latitude: 44.54; Longitude: 11.02 | |
| Latitudinal grid length | 5 km | |
| Longitudinal grid length | 5 km | |
| Model output time step | Daily | |
| Nesting | Two-way | |
| Land surface model | Noah-MP | |
| Simulation period | 1/1/2006 – 31/12/2015 | |
| Spin-up period | 1/1/2005 – 31/12/2005 | |
| Microphysics | New Thompson | (Thompson et al., 2008) |
| Shortwave radiation | Dudhia scheme | (Dudhia, 1989) |
| Longwave radiation | Rapid Radiative Transfer Model | (Mlawer et al., 1997) |
| Surface layer | Revised MM5 | (Jiménez et al., 2012b;Chen and Dudhia, 2001) |
| Planetary boundary layer | Yonsei University method | (Hong et al., 2006b) |
| Cumulus Parameterization | Kain-Fritsch (new Eta) scheme | (Kain, 2004a) |







**Table 2.** The relationship between the percentage of grids, and the cross-correlation.

| Cross-correlation ($r$) | Percentage of grids (%) |
|:---:|:---:|
| 0.5 | 85 |
| 0.6 | 78 |
| 0.7 | 70 |
| 0.8 | 52 |
| 0.9 | 15 |
| 0.95 | 3 |






**Table 3.** The number of components and grids to reach % variance threshold (based on the
PCA method and the Elbow curve method).

| Variance (%) | Components | Number of grids |
|---|---|---|
| 70.0 | 1 | 6 |
| 75.0 | 1 | 7 |
| 80.0 | 2 | 11 |
| 85.0 | 2 | 21 |
| 90.0 | 3 | 50 |
| 92.5 | 3 | 94 |
| 95.0 | 3 | 184 |
| 97.5 | 3 | 367 |







**Table 4.** *NSE* and correlation *r* performance of CA_Med and CA_Max.

| Variance | CA_Max_Mean | | CA_Med_Mean | | CA_Max_STD | | CA_Med_STD | |
|---|---|---|---|---|---|---|---|---|
| | NSE | r | NSE | r | NSE | r | NSE | r |
| 70.0 | 0.831 | 0.978 | 0.949 | 0.985 | 0.601 | 0.834 | 0.716 | 0.876 |
| 75.0 | 0.851 | 0.984 | 0.978 | 0.993 | 0.778 | 0.887 | 0.746 | 0.870 |
| 80.0 | 0.894 | 0.990 | 0.991 | 0.996 | 0.867 | 0.945 | 0.901 | 0.951 |
| 85.0 | 0.976 | 0.997 | 0.991 | 0.998 | 0.926 | 0.967 | 0.930 | 0.976 |
| **90.0** | 0.988 | 0.998 | **0.995** | **0.999** | 0.963 | 0.986 | **0.973** | **0.990** |
| 92.5 | 0.997 | 0.998 | 0.990 | 0.999 | 0.969 | 0.989 | 0.960 | 0.992 |
| 95.0 | 0.994 | 0.999 | 0.985 | 0.999 | 0.932 | 0.990 | 0.914 | 0.986 |
| 97.5 | 0.988 | 1.000 | 0.983 | 1.000 | 0.910 | 0.986 | 0.895 | 0.982 |











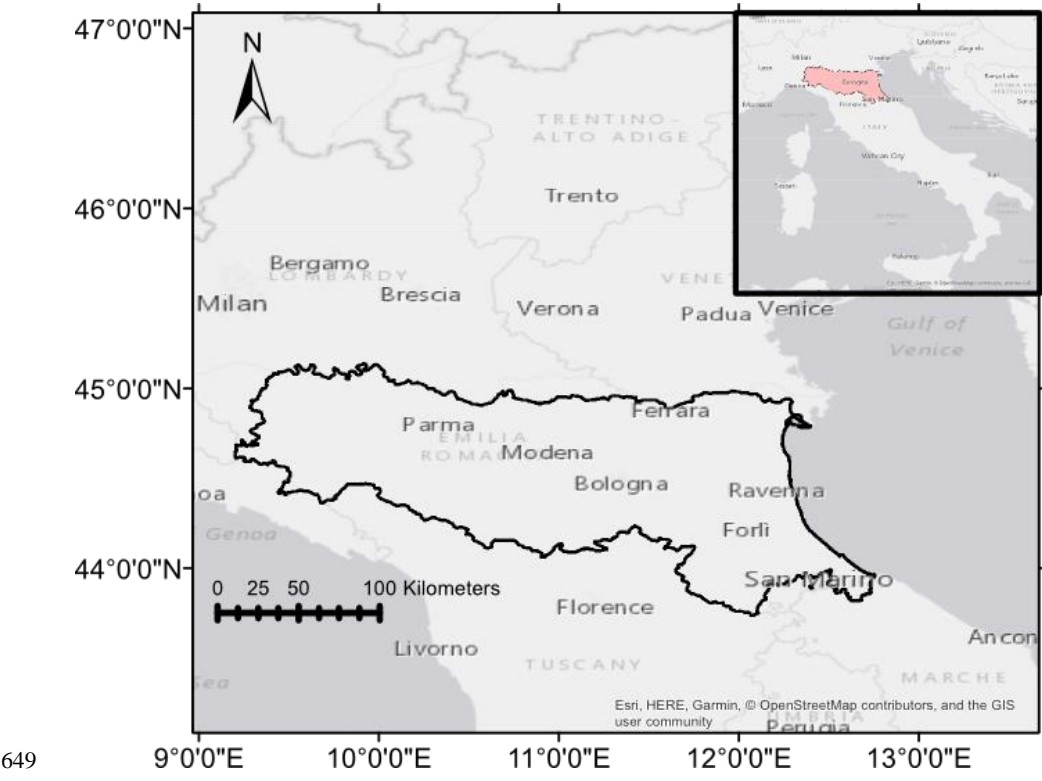

**Figure 1.** The geographical map of the Emilia Romagna region.





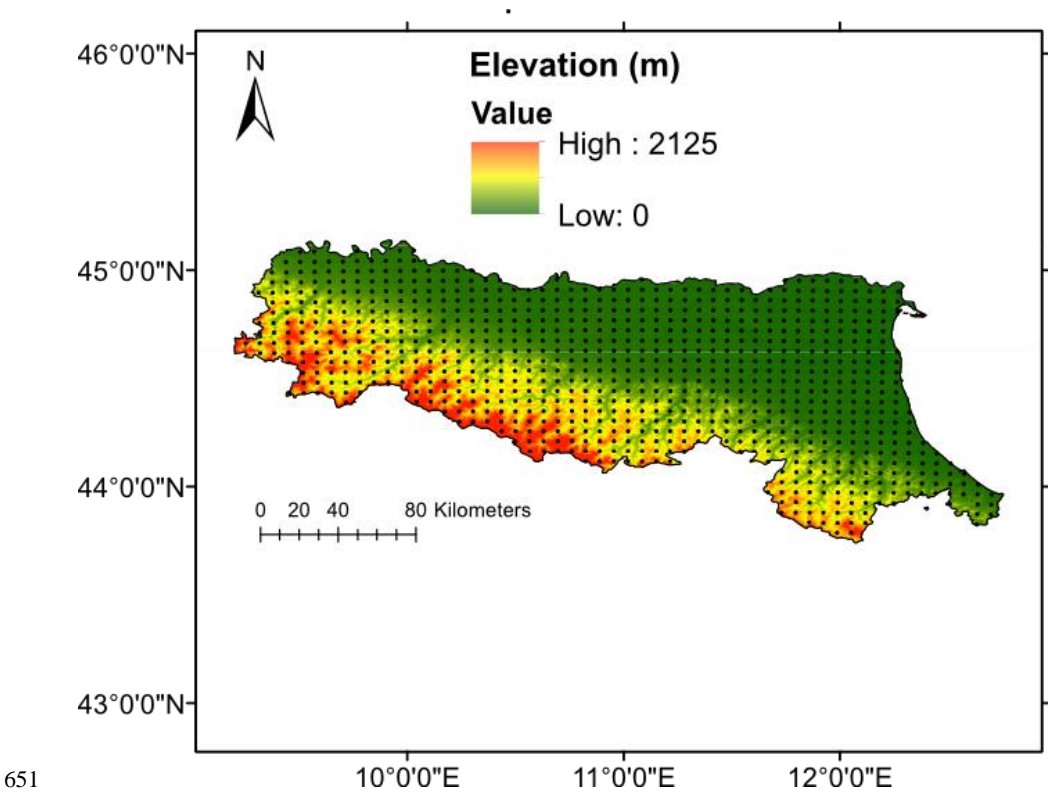



**Figure 2.** WRF grids used in the analysis, with DEM map in the background.



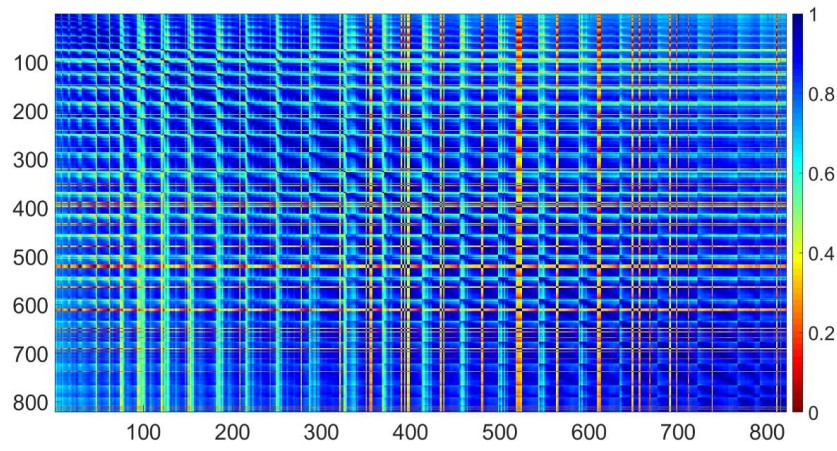


**Figure 3.** Cross correlation matrix for the whole catchment.




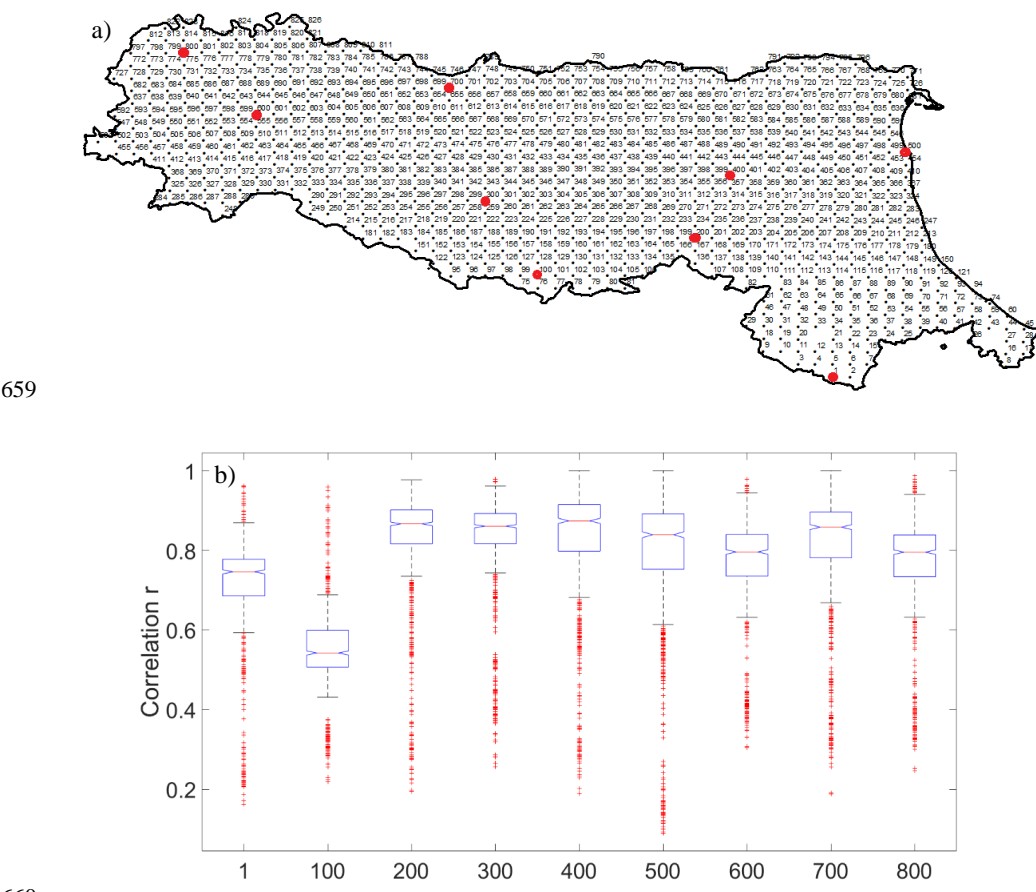



**Figure 4.** a) WRF grid number; b) correlation boxplot for the selected grids as highlighted in
red in a). For the boxplot, it shows the minimum, maximum, 0.25, 0.50, and 0.75 percentiles
and outliers (red cross).














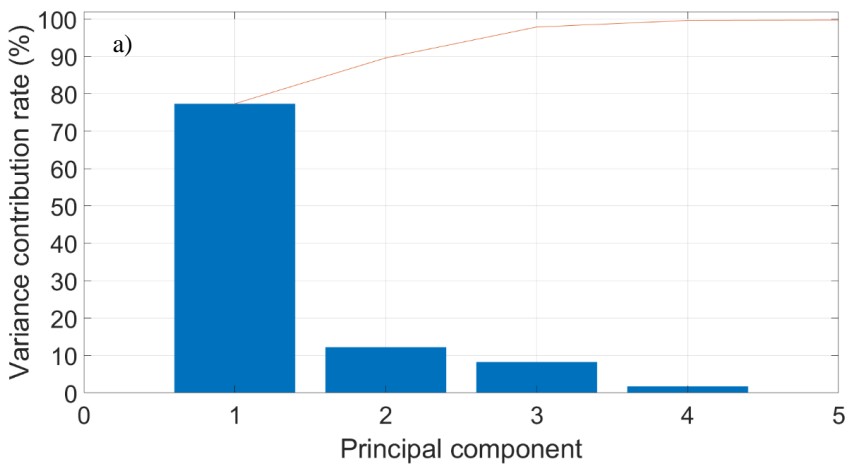


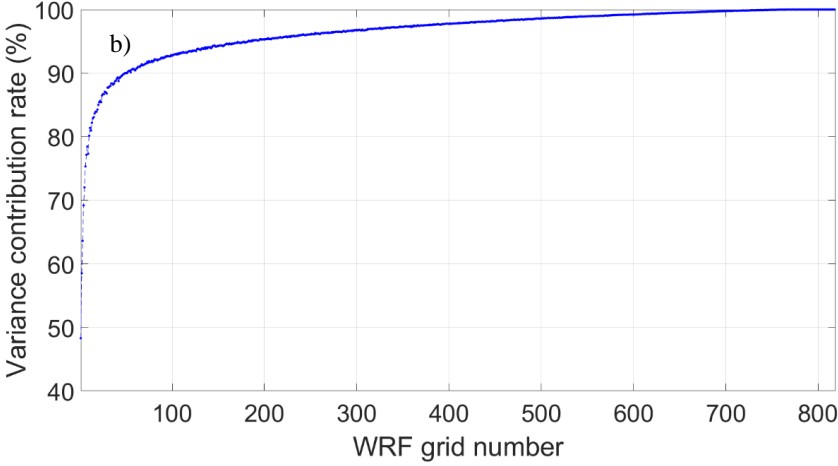


**Figure 5.** a) PCA analysis; b) Elbow curve.





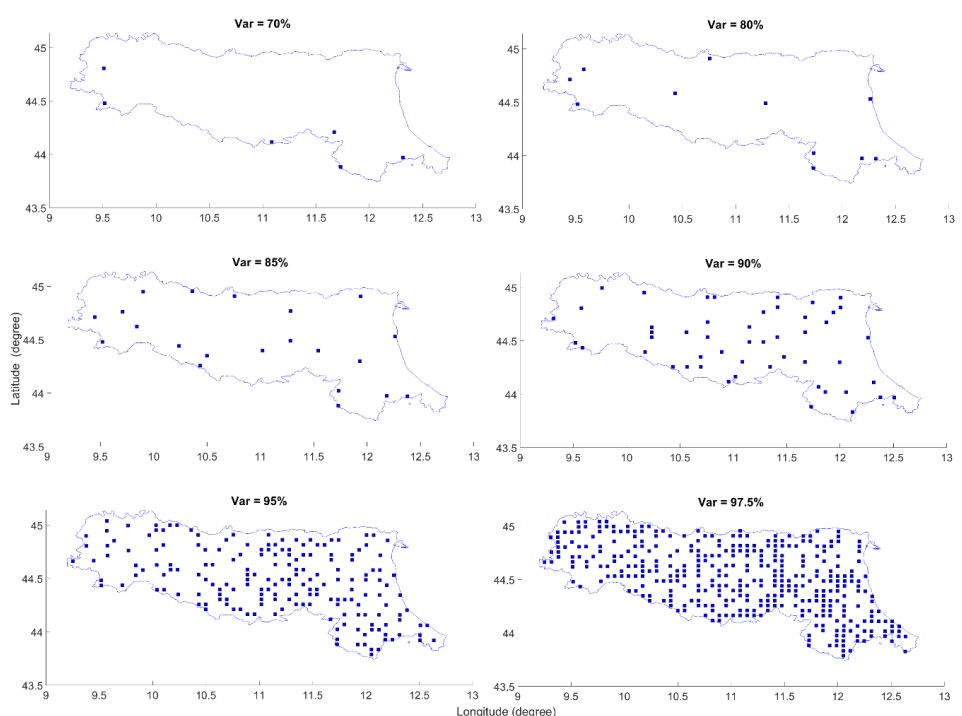


**Figure 6.** Designed soil moisture sensor locations, based on CA-Max.




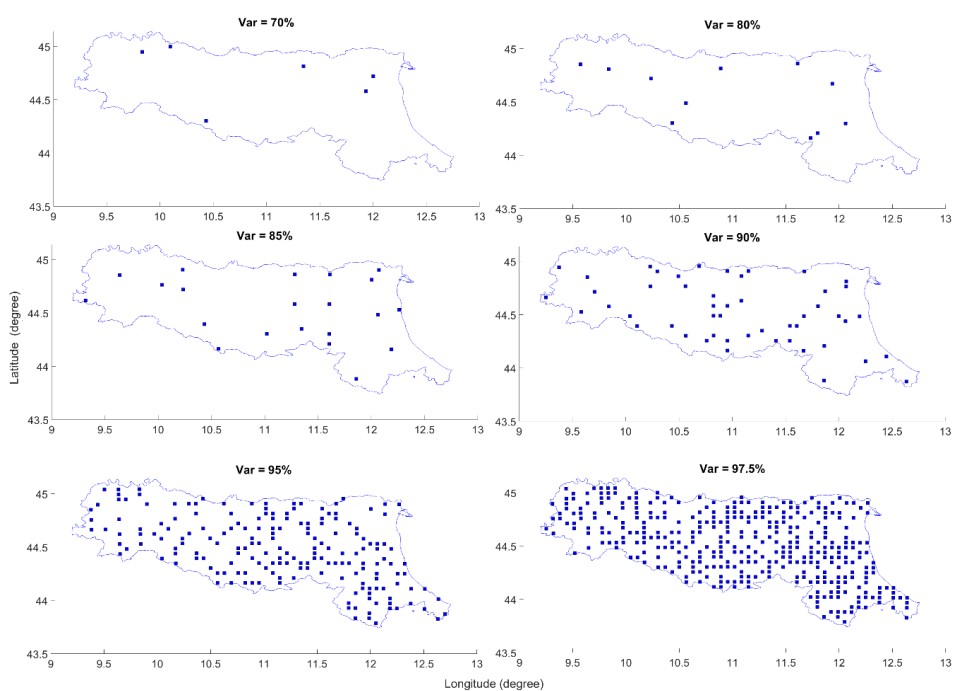


**Figure 7.** Designed soil moisture sensor locations, based on CA-Med.





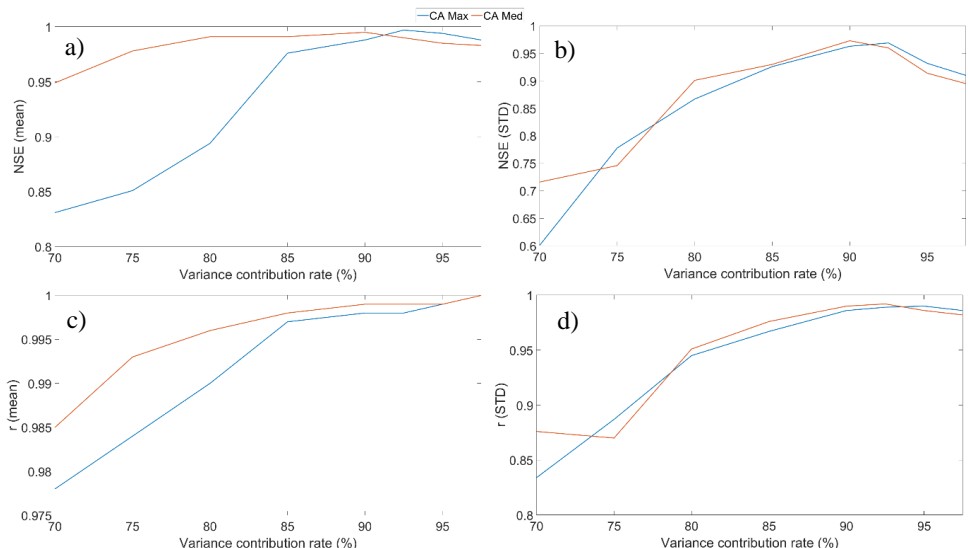

**Figure 8.** *NSE* and *r* plots: a) *NSE* performance based on the areal mean soil moisture, b) *NSE* performance based on the areal standard deviation soil moisture (STD), c) *r* performance based on the areal mean soil moisture, d) *r* performance based on the areal standard deviation soil moisture.



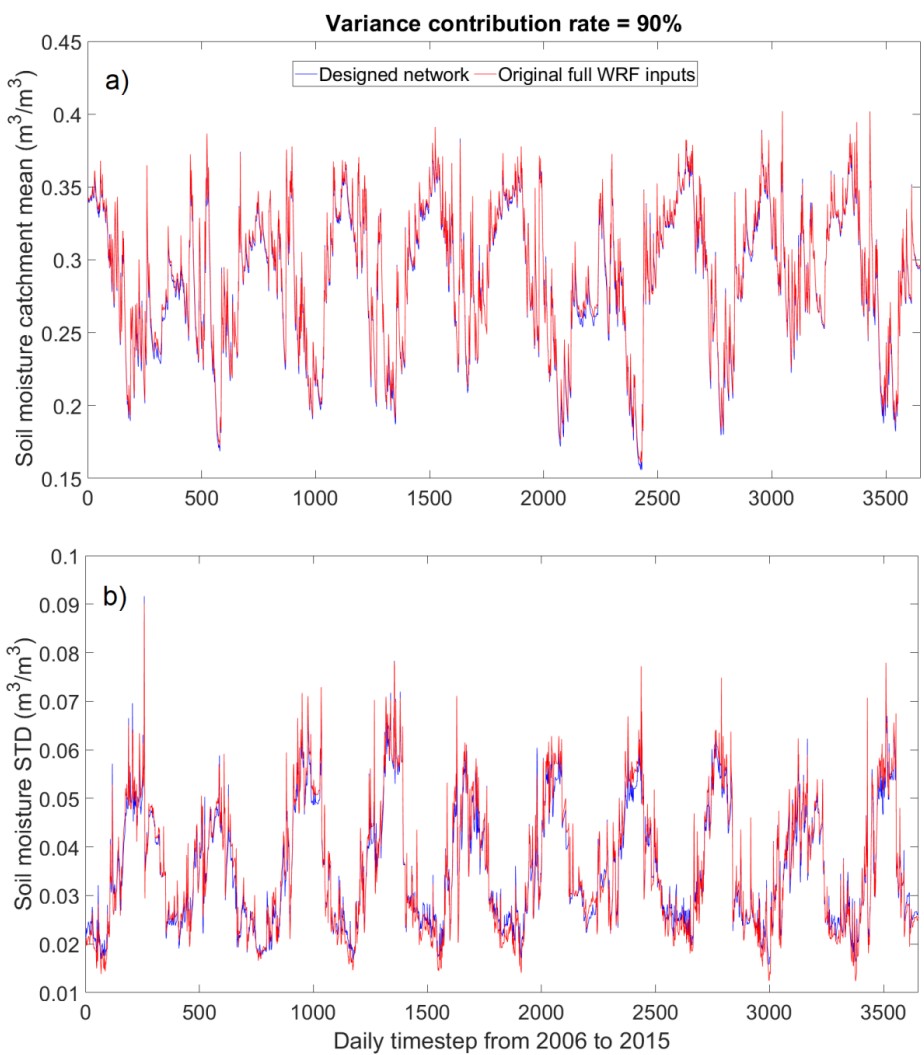

**Figure 9.** a) The areal mean soil moisture of the designed and the WRF's full-input networks,
b) the areal soil moisture standard deviation of the designed and the WRF's full-input networks.



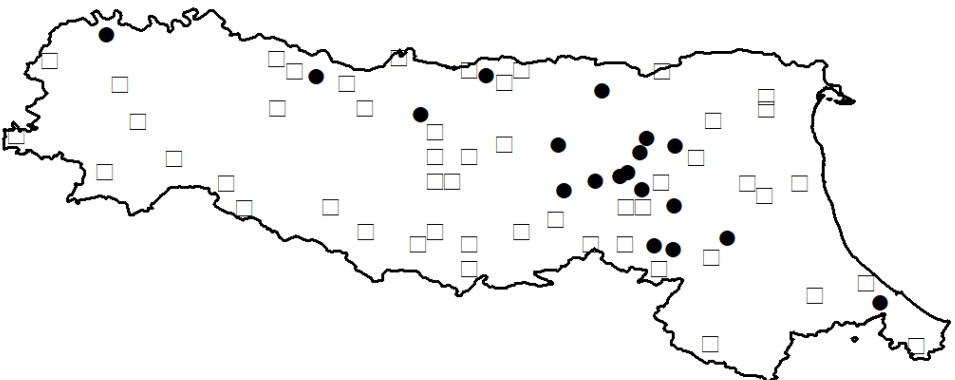


**Figure 10.** Comparison between the existing and the designed soil moisture networks.


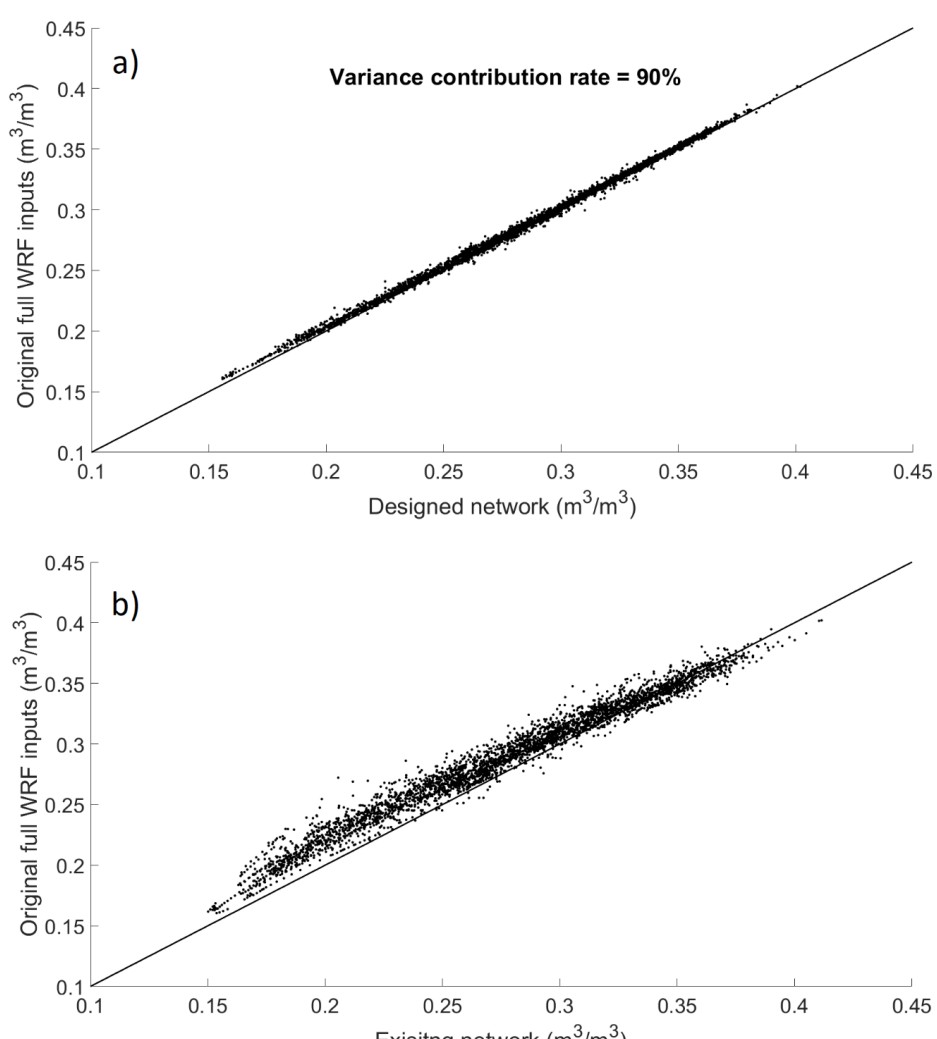


**Figure 11.** Scatterplots for areal mean soil moisture: a) WRF full grid inputs against the proposed network (*NSE* = 0.995, *r* = 0.998); b) WRF full grid inputs against the existing in-situ network (*NSE* = 0.889, *r* = 0.987).



