# Peer review of "Soil Moisture Sensor Network Design for Hydrological Applications"

_Hydrology and Earth System Sciences, 2020_

## Referee Comment (RC1) · Anonymous Referee #1 · 12 Feb 2020

The authors present a useful study on soil moisture network design for use in catchment modelling studies. The approach is fairly novel although the authors fail to cite some key parts of the literature on soil moisture networks and spatial scaling. Much work has been done through various NASA soil moisture campaigns (SMEX, SMAPVEX, etc.) that have provided a wealth of publications. In addition, the approaches of temporal stability analysis and Empirical Orthogonal Functions have been used in the soil moisture community for many years to address key issues on scaling and design. The paper is generally well written but needs greater tie in to the existing soil moisture community literature before publication. Additional grammar editing is also needed, see minor comments for some edits.

Major Comments

1. The authors fail to identify the depth of soil moisture information used in the WRF NoahMP analysis! I assume this is the rootzone integrated product from NoahMP 4 soil layers? Given the large difference between surface and rootzone soil moisture dynamics in space and time I would suggest doing the analysis for both a surface and rootzone layers. I suspect the network design will be different depending on which depth is of interest. Also what are the depths of sensors for the in-situ network?

2. The authors fail to cite key soil moisture techniques for scaling (see Crow 2012 for general review paper, Mohanty 2001 and Famiglietti 2008), mainly the approaches of Temporal Stability Analysis (TSA, Vachaud 1985) and Empirical Orthogonal Functions (EOF, Perry and Niemann 2007). I also suspect the EOF approach and comparison with environmental covariates is very similar to the PCA and CA approach used here (see Wang 2017). Moreover, I am concerned about the influence of temporal variation in the covariance analysis here. While EOF is very similar to PCA, EOF notably splits the variance into time and space components. Wang 2017 find the analysis of the regional NE Mesonet soil sensors that the spatial variability is dominated by the first EOF/PCA and that EOF is highly correlated to clay/sand fraction. I suspect the EOF approach would be enlightening here to show that topographic relief and/or soil texture dominated the 1st EOF from the NoahMP output. Also the map of 1st EOF coefficients will act as a form of spatial clustering analysis. I suspect that the alluvial plains will have similar EOF coefficients, similar to what was found with the clustering analysis here?

Vachaud, G., A. P. Desilans, P. Balabanis, and M. Vauclin (1985), Temporal stability of spatially measured soil-water probability density-function, Soil Sci. Soc. Am. J., 49(4), 822-828.

Perry, M. A., and J. D. Niemann (2007), Analysis and estimation of soil moisture at the catchment scale using EOFs, Journal of Hydrology, 334(3-4), 388-404. doi:10.1016/j.jhydrol.2006.10.014.

[Figure]

Famiglietti, J. S., D. R. Ryu, A. A. Berg, M. Rodell, and T. J. Jackson (2008), Field observations of soil moisture variability across scales, Water Resources Research, 44(1), 16. doi:W01423 10.1029/2006wr005804.

Mohanty, B. P., and T. H. Skaggs (2001), Spatio-temporal evolution and time-stable characteristics of soil moisture within remote sensing footprints with varying soil, slope, and vegetation, Adv. Water Resour., 24(9-10), 1051-1067. doi:10.1016/s0309-1708(01)00034-3.

Crow, W. T., A. A. Berg, M. H. Cosh, A. Loew, B. P. Mohanty, R. Panciera, P. de Rosnay, D. Ryu, and J. P. Walker (2012), Upscaling Sparse Ground-Based Soil Moisture Observations For The Validation Of Coarse-Resolution Satellite Soil Moisture Products, Rev. Geophys., 50. doi:10.1029/2011rg000372.

Wang, T., T. E. Franz, R. Li, J. You, M. D. Shulski, and C. Ray (2017), Evaluating climate and soil effects on regional soil moisture spatial variability using EOFs, Water Resources Research, 53. doi:10.1002/2017WR020642.

3. The scaling of a point sensor to a 5 km grid is not trivial (see Crow 2012). Additional geophysical approaches like GNSS or CRNS can provide integrated soil moisture data at a scale of tens to hundreds of meters as opposed to having a network of point sensors. The CRNS has been implemented with the COSMOS and COSMOSUK networks (Zreda 2012, Evans 2016). COSMOSUK network moving towards integration with operational weather forecasts. CRNS better suited for use in complex terrain and may be a good option to use for a national network as compared to in-situ point sensors. Cost over 10 years probably similar to point sensors with increased maintenance replacement etc. The sensor networks would provide different data for different purposes.

Zreda, M., W. J. Shuttleworth, X. Xeng, C. Zweck, D. Desilets, T. E. Franz, and R. Rosolem (2012), COSMOS: The COsmic-ray Soil Moisture Observing System, Hydrology and Earth System Sciences, 16, 4079-4099. doi:10.5194/hess-16-1-2012.

Evans, J. G., H. C. Ward, J. R. Blake, E. J. Hewitt, R. Morrison, M. Fry, L. A. Ball, L. C. Doughty, J. W. Libre, O. E. Hitt, D. Rylett, R. J. Ellis, A. C. Warwick, M. Brooks, M. A. Parkes, G. M. H. Wright, A. C. Singer, D. B. Boorman, and A. Jenkins (2016), Soil water content in southern England derived from a cosmic-ray soil moisture observing system - COSMOS-UK, Hydrological Processes, 30(26), 4987-4999. doi:10.1002/hyp.10929.

4. Much work has been done on soil moisture network design and implementation. See the NSMN for USA and ISMN databases for globe. These networks and efforts should be better acknowledged.

http://nationalsoilmoisture.com/ and papers by S. Quiring.

Dorigo, W. A., A. Xaver, M. Vreugdenhil, A. Gruber, A. Hegyiova, A. D. Sanchis-Dufau, D. Zamojski, C. Cordes, W. Wagner, and M. Drusch (2013), Global Automated Quality Control of In Situ Soil Moisture Data from the International Soil Moisture Network, Vadose Zone Journal, 12(3), 21. doi:10.2136/vzj2012.0097.

5. What do the CA clusters look like, that is are they nonconvex? How did you chose the optimal number of clusters (see Amiri 2019)? Please add more information on the CA approach used here.

Amiri, S., B. S. Clarke, J. L. Clarke, and H. Koepke (2019), A General Hybrid Clustering Technique, Journal of Computational and Graphical Statistics, 28(3), 540-551. doi:10.1080/10618600.2018.1546593. Minor comments

L 14. "variable in hydrological"

L 53. "for hydrological research"

L54. Large space? Sentence needs editing.

L62. NSMN and ISMN address are key sources of soil moisture information. In USA state Mesonets are designed to fill such gaps. See http://nationalsoilmoisture.com/ and OK Mesonet, NE Mesonet, SCAN, CRN for some networks available etc.

71. "of soil moisture"

72. Would disagree. TSA and EOF approaches have been used for such purposes.

Vachaud, G., A. P. Desilans, P. Balabanis, and M. Vauclin (1985), Temporal stability of spatially measured soil-water probability density-function, Soil Sci. Soc. Am. J., 49(4), 822-828.

Perry, M. A., and J. D. Niemann (2007), Analysis and estimation of soil moisture at the catchment scale using EOFs, Journal of Hydrology, 334(3-4), 388-404. doi:10.1016/j.jhydrol.2006.10.014.

Wang, T., T. E. Franz, R. Li, J. You, M. D. Shulski, and C. Ray (2017), Evaluating climate and soil effects on regional soil moisture spatial variability using EOFs, Water Resources Research, 53. doi:10.1002/2017WR020642.

Crow, W. T., A. A. Berg, M. H. Cosh, A. Loew, B. P. Mohanty, R. Panciera, P. de Rosnay, D. Ryu, and J. P. Walker (2012), Upscaling Sparse Ground-Based Soil Moisture Observations For The Validation Of Coarse-Resolution Satellite Soil Moisture Products, Rev. Geophys., 50. doi:10.1029/2011rg000372.

L108. Sentence is awkward, please revise.

L114. What are soil depths for model? Surface or rootzone?

L 180. section 3.2.1. How do you deal with temporal component of variation in PCA? EOF splits temporal and spatial components to identify dominant spatial structures.

Perry, M. A., and J. D. Niemann (2007), Analysis and estimation of soil moisture at the catchment scale using EOFs, Journal of Hydrology, 334(3-4), 388-404. doi:10.1016/j.jhydrol.2006.10.014.

L 217. How do you deal with nonconvex clusters and selection of number of clusters (Amiri 2019)? PCA + CA seems similar to EOF approach used by others.

Amiri, S., B. S. Clarke, J. L. Clarke, and H. Koepke (2019), A General Hybrid Clustering Technique, Journal of Computational and Graphical Statistics, 28(3), 540-551. doi:10.1080/10618600.2018.1546593.

L257. KGE criteria has been shown to be superior to NSE (Gupta 2009). KGE uses correlation, bias in mean and standard deviation. Here you use both NSE and correlation, why not switch to KGE for simplicity?

Gupta, H. V., H. Kling, K. K. Yilmaz, and G. F. Martinez (2009), Decomposition of the mean squared error and NSE performance criteria: Implications for improving hydrological modelling, Journal of Hydrology, 377(1-2), 80-91. doi:10.1016/j.jhydrol.2009.08.003.

L411-413. I would disagree. See comments above about work on soil moisture scaling and implications to network design, like NSMN or USA state Mesonets.

L458. CRNS good option for long-term deployment in complex terrain.

Zreda, M., W. J. Shuttleworth, X. Xeng, C. Zweck, D. Desilets, T. E. Franz, and R. Rosolem (2012), COSMOS: The COsmic-ray Soil Moisture Observing System, Hydrology and Earth System Sciences, 16, 4079-4099. doi:10.5194/hess-16-1-2012.

Evans, J. G., H. C. Ward, J. R. Blake, E. J. Hewitt, R. Morrison, M. Fry, L. A. Ball, L. C. Doughty, J. W. Libre, O. E. Hitt, D. Rylett, R. J. Ellis, A. C. Warwick, M. Brooks, M. A. Parkes, G. M. H. Wright, A. C. Singer, D. B. Boorman, and A. Jenkins (2016), Soil water content in southern England derived from a cosmic-ray soil moisture observing system - COSMOS-UK, Hydrological Processes, 30(26), 4987-4999. doi:10.1002/hyp.10929.

---

## Referee Comment (RC2) · Anonymous Referee #2 · 6 Mar 2020

This manuscript address an important problem in hydrology, the optimal design of soil moisture monitoring networks. The approach utilizes principal components analysis and cluster analysis informed by gridded data from the WRF weather forecasting model. The general approach shows good potential, although no observed data were available for testing, only the WRF soil moisture outputs. I have two primary concerns with the manuscript.

First, the approach is unclear. In particular, the relationships between the number of principal components, the number of clusters, and the number of station locations need to be more explicitly described. The assumptions related to these relationships need to be stated and justified.

Second, a major source of uncertainty about the success of the method needs to be

added to the text. The method implicitly assumes that a soil moisture station placed inside a 5-km grid cell will perfectly represent the mean soil moisture condition for that grid cell. Of course, in reality it will not do so. The scale mismatch between the footprint of an in situ soil moisture station and the 5-km data set used here would be expected to degrade the performance of the resulting network. The uncertainty introduced by this scale mismatch may be quite large and cannot be quantified by the data in the manuscript. This issue needs to be discussed in the text.

I have included 55 specific comments, edits, and questions in a pdf version of the manuscript attached with this review.

Please also note the supplement to this comment:
https://www.hydrol-earth-syst-sci-discuss.net/hess-2020-24/hess-2020-24-RC2-supplement.pdf

―――――――――――――――――

**Supplement:**

[revised manuscript text omitted]

---

## Referee Comment (RC3) · Anonymous Referee #3 · 10 Mar 2020

General comments to the authors The proposed approach to identify the optimal number and location of a limited number of soil moisture sensors is sound and leverages well-known statistical techniques such as Principal Component Analysis and K-means clustering analysis. Clearly the research community is active in this topic and approaches of this type will be valuable to the scientific community, state and federal agencies, and watershed managers. While the approach is sound, it has two important cons in my opinion: 1) The approach requires a substantial amount of available information about the variable in question to decide the architecture of the network. This somewhat conflicts with the idea of deploying a new network, which is trying to resolve the problem of no having soil moisture data available. The second drawback is that watershed managers will likely not spend substantial amount of time running complex

simulations to generate a dataset to inform the proposed approach unless they are assisted by a group of scientists. I think the manuscript deserves publication. Below are some suggestions and questions to allow the authors improve the manuscript.

Specific comments to the authors Line 49: Remove comma between "probe" and "and Time.." Line 50: Compared to other technologies such as neutron scattering and the gravimetric method, sensors relying on electromagnetic principles are probably not that old, particularly from the point of view of automated systems. They are about 30 years old, but not sure whether it classifies as one of the oldest. Line 51: When calibrated, these sensors can be accurate. Otherwise measurements can have substantial bias. Line 54: Economic considerations about what aspect? Please clarify. Line 95: I suggest being more explicit and write "Po river plain and surrounding hilly areas" Line 97-98: Please define acronym "ESA CCI" Line 152: Please, consider a modification along these lines "5-minute angle soil database" or "5-minutes geographic resolution soil database" Line 157: Please use square brackets when nesting parentheses. Line 197: Remove "the" from "since the eigenvectors of the X" What is the depth of the soil moisture sensing by the current stations? How will this approach handle multiple sensing depths? It seems that the final network configuration may be different for different soil layers. In Figure 9 it is unclear what is the number of stations considered and the method (CA-max, CA-median) employed to create the "designed network". Please provide more details in the figure caption. Do the current soil moisture monitoring stations only measure soil moisture? Do they observe other hydrological or meteorological variables that need to be taken into account at the time of designing a new network? To account for the network designed in probabilistic terms, have the authors considered using a probabilistic clustering method such as the Fuzzy C-means approach? Does the proposed approach account for the layout of existing monitoring stations? By looking at Figure 10, it seems that a complete re-arrangement of stations is required, which may conflict with existing agency resources and manpower. Judging from Figure 11 it seems that the whole effort of re-designing the network will remove some existing bias, but may not lead to a substantial improvement. Can the authors provide more details

on the logic behind the location of existing stations? It seems that the CA-max proposed clustering technique tends to initially place stations at or near the edges of the watershed. While this is the result of the proposed objective method, there are reasons why this might not be ideal. The CA-median seems to be less sensitive to this. How is the PCA superior to defining the number of clusters using something like the Silhouette method? How is the k-means part of this method different than variance minimization methods based on correlograms or semi-variograms? How does this approach include or handle the changing land use/land cover? Is it How can this method be applied to smaller catchment areas where a single pixel from remote sensing sources or distributed models is equal or even larger than the entire watershed?

---

## Author Comment (AC1) · 7 Apr 2020

**Replies to Reviewer 1**

The authors present a useful study on soil moisture network design for use in catchment modelling studies. The approach is fairly novel although the authors fail to cite some key parts of the literature on soil moisture networks and spatial scaling. Much work has been done through various NASA soil moisture campaigns (SMEX, SMAPVEX, etc.) that have provided a wealth of publications. In addition, the approaches of temporal stability analysis and Empirical Orthogonal Functions have been used in the soil moisture community for many years to address key issues on scaling and design. The paper is generally well written but needs greater tie in to the existing soil moisture community literature before publication. Additional grammar editing is also needed, see minor comments for some edits.

Reply: We thank the reviewer in acknowledging the usefulness and novelty of the proposed soil moisture network design scheme for catchment modelling studies.

The NASA soil moisture campaigns and other similar projects are mainly focused on satellite soil moisture evaluations and algorithm improvements, so the in-situ sensors are purposely designed to best match satellite's observational footprint. Therefore their target is different to this study's which is focused on large catchment scale application. However, we agree with the reviewer that these researches should be cited and described. We will add them in the updated manuscript. Regarding the different statistical approaches mentioned by the reviewer, they will also be added in the manuscript.

Major Comments

1.  The authors fail to identify the depth of soil moisture information used in the WRF NoahMP analysis! I assume this is the root zone integrated product from NoahMP 4 soil layers? Given the large difference between surface and rootzone soil moisture dynamics in space and time I would suggest doing the analysis for both a surface and rootzone layers. I suspect the network design will be different depending on which depth is of interest. Also what are the depths of sensors for the in-situ network?

Reply: The soil depth used in this study is the surface layer from the WRF NoahMP (top 10 cm). We agree with the reviewer that the result of the network design could be different depending on which depth is used for the analysis. The sensor depths in the in-situ network varies, but the majority are centred at 10 cm, 25 cm, 45 cm and 70 cm.  The NoahMP provides soil moisture centred at 10 cm, 25 cm, 70 cm, and 150 cm. We will add the analysis of using the root zone soil moisture in the updated manuscript as suggested by the reviewer. And based on the common depth between the in-situ sensors and the NoahMP, 25 and 70 cm will be integrated to calculate the overall root zone soil moisture.

2.  The authors fail to cite key soil moisture techniques for scaling (see Crow 2012 for general review paper, Mohanty 2001 and Famiglietti 2008), mainly the approaches of Temporal Stability Analysis (TSA, Vachaud 1985) and Empirical Orthogonal Functions (EOF, Perry and Niemann 2007). I also suspect the EOF approach and comparison with environmental covariates is very similar to the PCA and CA approach used here (see Wang 2017). Moreover, I am concerned about the influence of temporal variation in the covariance analysis here. While EOF is very similar to PCA, EOF notably splits the variance into time and space components. Wang 2017 find the analysis of the regional NE Mesonet soil

sensors that the spatial variability is dominated by the first EOF/PCA and that EOF is highly correlated to clay/sand fraction. I suspect the EOF approach would be enlightening here to show that topographic relief and/or soil texture dominated the 1st EOF from the NoahMP output. Also the map of 1st EOF coefficients will act as a form of spatial clustering analysis. I suspect that the alluvial plains will have similar EOF coefficients, similar to what was found with the clustering analysis here?

Vachaud, G., A. P. Desilans, P. Balabanis, and M. Vauclin (1985), Temporal stability of spatially measured soil-water probability density-function, Soil Sci. Soc. Am. J., 49(4), 822-828.

Perry, M. A., and J. D. Niemann (2007), Analysis and estimation of soil moisture at the catchment scale using EOFs, Journal of Hydrology, 334(3-4), 388-404. doi:10.1016/j.jhydrol.2006.10.014.

Famiglietti, J. S., D. R. Ryu, A. A. Berg, M. Rodell, and T. J. Jackson (2008), Field observations of soil moisture variability across scales, Water Resources Research, 44(1), 16. doi:W01423 10.1029/2006wr005804.

Mohanty, B. P., and T. H. Skaggs (2001), Spatio-temporal evolution and time-stable characteristics of soil moisture within remote sensing footprints with varying soil, slope, and vegetation, Adv. Water Resour., 24(9-10), 1051-1067. doi:10.1016/s0309-1708(01)00034-3.

Crow, W. T., A. A. Berg, M. H. Cosh, A. Loew, B. P. Mohanty, R. Panciera, P. de Rosnay, D. Ryu, and J. P. Walker (2012), Upscaling Sparse Ground-Based Soil Moisture Observations For The Validation Of Coarse-Resolution Satellite Soil Moisture Products, Rev.Geophys., 50. doi:10.1029/2011rg000372.

Wang, T., T. E. Franz, R. Li, J. You, M. D. Shulski, and C. Ray (2017), Evaluating climate and soil effects on regional soil moisture spatial variability using EOFs, Water Resources Research, 53. doi:10.1002/2017WR020642.

Reply: We agree with the reviewer that PCA/CA combination might not be the only approach that could be explored for the soil moisture network design. The mentioned studies will be added and described in the updated manuscript.

Regarding the temporal variation factor, it should be noted that the information we used for the PCA/CA is based on the soil moisture temporal variations (e.g., the 10-year time series data), so that areas following similar soil moisture temporal variations can be identified and only one sensor will be needed to represent them. Location information is not used for the PCA/CA analysis. However, due to the influence of local characteristics, the resultant clusters should more or less reflect the geographical feature. This information was not included in the manuscript, which will be updated. We had plotted the clusters in the following figure. It can be seen that most of the clusters are geographically connected. Whilst k-means has issues dealing with nonconvex clusters and geographically we might have nonconvex shaped clusters, but in terms of the result, k-means indeed is very useful for the soil moisture network design. We have tried EOF, however, we found it very difficult in dealing with a large array of datasets (828 points, and each with 3652 datasets), which was therefore not considered in this study.

[Figure]

3. The scaling of a point sensor to a 5 km grid is not trivial (see Crow 2012). Additional geophysical approaches like GNSS or CRNS can provide integrated soil moisture data at a scale of tens to hundreds of meters as opposed to having a network of point sensors. The CRNS has been implemented with the COSMOS and COSMOSUK networks (Zreda 2012, Evans 2016). COSMOSUK network moving towards integration with operational weather forecasts. CRNS better suited for use in complex terrain and may be a good option to use for a national network as compared to in-situ point sensors. Cost over 10 years probably similar to point sensors with increased maintenance replacement etc. The sensor networks would provide different data for different purposes.

Zreda, M., W. J. Shuttleworth, X. Xeng, C. Zweck, D. Desilets, T. E. Franz, and R. Rosolem (2012), COSMOS: The COsmic-ray Soil Moisture Observing System, Hydrology and Earth System Sciences, 16, 4079-4099. doi:10.5194/hess-16-1-2012.

Evans, J. G., H. C. Ward, J. R. Blake, E. J. Hewitt, R. Morrison, M. Fry, L. A. Ball, L. C. Doughty, J. W. Libre, O. E. Hitt, D. Rylett, R. J. Ellis, A. C. Warwick, M. Brooks, M. A. Parkes, G. M. H. Wright, A. C. Singer, D. B. Boorman, and A. Jenkins (2016), Soil water content in southern England derived from a cosmic-ray soil moisture observing system - COSMOS-UK, Hydrological Processes, 30(26), 4987-4999. doi:10.1002/hyp.10929.

Reply: Indeed. We agree with the reviewer, that the scale mismatch between the footprint of a point-based in-situ soil moisture station and a 5-km model gird would be expected to degrade the performance of the resulting network. The advanced soil moisture sensors based on GNSS and COSMIC-RAY could provide alternative solutions to overcome the mismatch problem. We will add a discussion in the updated manuscript.

4. Much work has been done on soil moisture network design and implementation. See the NSMN for USA and ISMN databases for globe. These networks and efforts should be better acknowledged.

http://nationalsoilmoisture.com/ and papers by S. Quiring. Dorigo, W. A., A. Xaver, M. Vreugdenhil, A. Gruber, A. Hegyiova, A. D. Sanchis-Dufau, D. Zamojski, C. Cordes, W.

Wagner, and M. Drusch (2013), Global Automated Quality Control of In Situ Soil Moisture Data from the International Soil Moisture Network, Vadose Zone Journal, 12(3), 21. doi:10.2136/vzj2012.0097.

Reply: Agreed. They will be added in the updated manuscript.

5. What do the CA clusters look like, that is are they nonconvex? How did you chose the optimal number of clusters (see Amiri 2019)? Please add more information on the CA approach used here.

Amiri, S., B. S. Clarke, J. L. Clarke, and H. Koepke (2019), A General Hybrid Clustering Technique, Journal of Computational and Graphical Statistics, 28(3), 540-551. doi:10.1080/10618600.2018.1546593.

Reply: Regarding the nonconvex clusters, please see the reply to comment 2.

On choosing the optimal number of clusters, the methodology part regarding the use of PCA, and CA in the existing manuscript is not very well structured. It will be reorganised and rewritten to improve its clarity. In essence, PCA is used for network redundancy analysis. Since the number of components from the PCA do not directly represent the physical number of grids, we propose to use the elbow method to find the corresponding number of grids. The elbow method is based on K-means clustering and looks at the variance contribution rate as a function of the number of grids. Generally, the required number of grids increases when the variance contribution rate increases. However, the growth rate is not constant that normally changes significantly at a critical point (threshold), which is used in this study as the desired rate for the soil moisture network design. And the corresponding number of clusters will be used. The threshold is found through visual recognition (Figure 5), and comparison of statistical performances of NSE and r (i.e., Table 3 and 4, Figure 8).

**Minor comments**

L 14. "variable in hydrological"

Reply: This will be updated.

L 53. "for hydrological research"

Reply: This will be updated.

L54. Large space? Sentence needs editing.

Reply: This will be updated.

L62. NSMN and ISMN address are key sources of soil moisture information. In USA state Mesonets are designed to fill such gaps. See http://nationalsoilmoisture.com/ and OK Mesonet, NE Mesonet, SCAN, CRN for some networks available etc.

Reply: Agreed. As to comment 4, this will be added in the updated manuscript.

L71. "of soil moisture"

Reply: This will be updated.

L72. Would disagree. TSA and EOF approaches have been used for such purposes.

Vachaud, G., A. P. Desilans, P. Balabanis, and M. Vauclin (1985), Temporal stability of spatially measured soil-water probability density-function, Soil Sci. Soc. Am. J., 49(4), 822-828.

Perry, M. A., and J. D. Niemann (2007), Analysis and estimation of soil moisture at the catchment scale using EOFs, Journal of Hydrology, 334(3-4), 388-404. doi:10.1016/j.jhydrol.2006.10.014.

Wang, T., T. E. Franz, R. Li, J. You, M. D. Shulski, and C. Ray (2017), Evaluating climate and soil effects on regional soil moisture spatial variability using EOFs, Water Resources Research, 53. doi:10.1002/2017WR020642.

Crow, W. T., A. A. Berg, M. H. Cosh, A. Loew, B. P. Mohanty, R. Panciera, P. de Rosnay, D. Ryu, and J. P. Walker (2012), Upscaling Sparse Ground-Based Soil Moisture Observations For The Validation Of Coarse-Resolution Satellite Soil Moisture Products, Rev. Geophys., 50. doi:10.1029/2011rg000372.
Reply: Agreed. As to comment 2. These existing studies will be added in the updated manuscript.

L108. Sentence is awkward, please revise.
Reply: This will be modified.

L114. What are soil depths for model? Surface or rootzone?
Reply: The surface soil moisture at 0-10m is used for the analysis. As to comment 1, additional analysis based on root zone soil moisture will be added in the updated manuscript.

L 180. section 3.2.1. How do you deal with temporal component of variation in PCA? EOF splits temporal and spatial components to identify dominant spatial structures.

Perry, M. A., and J. D. Niemann (2007), Analysis and estimation of soil moisture at the catchment scale using EOFs, Journal of Hydrology, 334(3-4), 388-404. doi:10.1016/j.jhydrol.2006.10.014.
Reply: Please see the reply to comment 2.

L 217. How do you deal with nonconvex clusters and selection of number of clusters (Amiri 2019)? PCA + CA seems similar to EOF approach used by others.

Amiri, S., B. S. Clarke, J. L. Clarke, and H. Koepke (2019), A General Hybrid Clustering Technique, Journal of Computational and Graphical Statistics, 28(3), 540-551. doi:10.1080/10618600.2018.1546593.
Reply: Please see the reply to comment 2 and 5.

L257. KGE criteria has been shown to be superior to NSE (Gupta 2009). KGE uses correlation, bias in mean and standard deviation. Here you use both NSE and correlation, why not switch to KGE for simplicity?

Gupta, H. V., H. Kling, K. K. Yilmaz, and G. F. Martinez (2009), Decomposition of the mean squared error and NSE performance criteria: Implications for improving hydrological modelling, Journal of Hydrology, 377(1-2), 80-91. doi:10.1016/j.jhydrol.2009.08.003.

Reply: We thank the reviewer on suggesting KGE for the performance assessment to replace the combinational use of NSE and r. We have found a recent paper written by Knoben, which compares NSE and KGE, it has suggested that "a strong case can be made for moving away from ad hoc use of aggregated efficiency metrics and towards a framework based on purpose-dependent evaluation metrics and benchmarks that allows for more robust model adequacy assessment". Although there is the advancement of using KGE over the NSE, it may still not be sufficient to use the KGE on its own. Therefore the combination of NSE and r will be kept in this paper.

Knoben, Wouter JM, Jim E. Freer, and Ross A. Woods. "Inherent benchmark or not? Comparing Nash–Sutcliffe and Kling–Gupta efficiency scores." Hydrology and Earth System Sciences 23.10 (2019): 4323-4331.

L411-413. I would disagree. See comments above about work on soil moisture scaling and implications to network design, like NSMN or USA state Mesonets.

Reply: This will be added, and the manuscript will be updated.

L458. CRNS good option for long-term deployment in complex terrain.

Zreda, M., W. J. Shuttleworth, X. Xeng, C. Zweck, D. Desilets, T. E. Franz, and R. Rosolem (2012), COSMOS: The COsmic-ray Soil Moisture Observing System, Hydrology and Earth System Sciences, 16, 4079-4099. doi:10.5194/hess-16-1-2012.

Evans, J. G., H. C. Ward, J. R. Blake, E. J. Hewitt, R. Morrison, M. Fry, L. A. Ball, L. C. Doughty, J. W. Libre, O. E. Hitt, D. Rylett, R. J. Ellis, A. C. Warwick, M. Brooks, M. A. Parkes, G. M. H. Wright, A. C. Singer, D. B. Boorman, and A. Jenkins (2016), Soil water content in southern England derived from a cosmic-ray soil moisture observing system - COSMOS-UK, Hydrological Processes, 30(26), 4987-4999. doi:10.1002/hyp.10929.

Reply: This will be added in the updated manuscript.

---

## Author Comment (AC2) · 7 Apr 2020

**Replies to Reviewer 2**

This manuscript address an important problem in hydrology, the optimal design of soil moisture monitoring networks. The approach utilizes principal components analysis and cluster analysis informed by gridded data from the WRF weather forecasting model. The general approach shows good potential, although no observed data were available for testing, only the WRF soil moisture outputs. I have two primary concerns with the manuscript.

Reply: We thank the reviewer in acknowledging the good potential of the paper. The two primary concerns are addressed as follows.

First, the approach is unclear. In particular, the relationships between the number of principal components, the number of clusters, and the number of station locations need to be more explicitly described. The assumptions related to these relationships need to be stated and justified.

Reply: We apologise that the methodology part needs further clarification. In essence, for the soil moisture network design, three main problems need to be tackled. The first is how redundant the network is, the second is how many soil moisture sensors are needed within a catchment, and finally where are the best locations to place them. To solve the first problem, the PCA is used to investigate the redundancy degree of the network (in relation to different variance contribution rates). For the latter two problems, the k-means cluster analysis is adopted (i.e., the elbow method for the determination of sensor number in accordance to the variance contribution rates), and CA-Med, CA-Max for finding the optimal sensor placements).

In the updated manuscript, the methodology section (i.e., Soil Moisture Network Design) will be reorganised and rewritten to avoid the concerned confusion to the readers.

Second, a major source of uncertainty about the success of the method needs to be added to the text. The method implicitly assumes that a soil moisture station placed inside a 5-km grid cell will perfectly represent the mean soil moisture condition for that grid cell. Of course, in reality it will not do so. The scale mismatch between the footprint of an in situ soil moisture station and the 5-km data set used here would be expected to degrade the performance of the resulting network. The uncertainty introduced by this scale mismatch may be quite large and cannot be quantified by the data in the manuscript. This issue needs to be discussed in the text.

Reply: We agree with the reviewer that a soil moisture station placed inside a 5-km grid cell cannot perfectly represent the mean soil moisture condition for that grid cell. Advanced soil moisture sensing technologies such as the Global Navigation Satellite Systems (GNSS) and the Cosmic-ray could provide alternative solutions over point-based sensors to reduce the mismatch impacts. In particular, COSMOSUK network is moving towards integration with operational weather forecasts, and Cosmic-ray is suitable in complex terrain and might be a good option to be used for national network as compared with in-situ point sensors (Cosmic-ray sensors are also more cost-effective [e.g., cost over 10 years is probably similar to point sensors with increased maintenance replacement etc.])

This discussion will be included in the updated manuscript.

I have included 55 specific comments, edits, and questions in a pdf version of the manuscript attached with this review.

Reply: We thank the reviewer for the detailed specific comments. They will all be addressed in the updated manuscript.

---

## Author Comment (AC3) · 7 Apr 2020

**Replies to Reviewer 3**

**General comments to the authors**
The proposed approach to identify the optimal number and location of a limited number of soil moisture sensors is sound and leverages well-known statistical techniques such as Principal Component Analysis and K-means clustering analysis. Clearly the research community is active in this topic and approaches of this type will be valuable to the scientific community, state and federal agencies, and watershed managers. While the approach is sound, it has two important cons in my opinion:
Reply: We thank the reviewer in acknowledging the proposed method as sound and valuable to the scientific community.
1) The approach requires a substantial amount of available information about the variable in question to decide the architecture of the network. This somewhat conflicts with the idea of deploying a new network, which is trying to resolve the problem of no having soil moisture data available.
Reply: Unlike methodologies proposed by other literature such as (Chaney et al., 2015) where a large number of in-situ characteristics datasets are required to build-up the soil moisture network (topography, land cover, soil properties), the methodology proposed in this study only requires soil moisture information from the WRF model (this will be emphasized in the discussion section). Since the WRF model can be run via globally freely available reanalysis datasets (e.g., ERA-5), it can be applied to any part of the world, even for areas where in-situ datasets are extremely sparse. Moreover, although WRF estimated soil moisture cannot represent the ground truth, they are ideal datasets to provide catchment and hydrometeorological characteristics, such as local climate, land cover, soil properties, topographies, which are the main drivers of local soil moisture heterogeneity. Therefore, we believe the proposed approach has the advantage over existing literature in soil moisture network design.

Chaney, N. W., Roundy, J. K., Herrera‐Estrada, J. E., and Wood, E. F.: High‐resolution modeling of the spatial heterogeneity of soil moisture: Applications in network design, Water Resour. Res., 51, 619-638, 2015.

2) The second drawback is that watershed managers will likely not spend substantial amount of time running complex simulations to generate a dataset to inform the proposed approach unless they are assisted by a group of scientists.
Reply: In the future, a decision support tool based on cloud computing could be developed, where WRF and the relevant data and code are ready to be applied anywhere in the world. In such ways, the watershed managers could focus on making optional choices instead of worrying about the background computational work.

I think the manuscript deserves publication. Below are some suggestions and questions to allow the authors improve the manuscript. Specific comments to the authors

Line 49: Remove comma between "probe" and "and Time.."
Reply: Agreed. This will be updated.

Line 50: Compared to other technologies such as neutron scattering and the gravimetric method, sensors relying on electromagnetic principles are probably not that old, particularly from the point of view of automated systems. They are about 30 years old, but not sure whether it classifies as one of the oldest.

Reply: Agreed. This will be updated.

Line 51: When calibrated, these sensors can be accurate. Otherwise measurements can have substantial bias.
Reply: Agreed. This will be updated.

Line 54: Economic considerations about what aspect? Please clarify.
Reply: This will be clarified.

Line 95: I suggest being more explicit and write "Po river plain and surrounding hilly areas"
Reply: Agreed. This will be updated as suggested.

Line 97-98: Please define acronym "ESA CCI"
Reply: The full name will be added.

Line 152: Please, consider a modification along these lines "5-minute angle soil database" or "5-minutes geographic resolution soil database"
Reply: It will be updated.

Line 157: Please use square brackets when nesting parentheses.
Reply: It will be updated.

Line 197: Remove "the" from "since the eigenvectors of the X" What is the depth of the soil moisture sensing by the current stations? How will this approach handle multiple sensing depths? It seems that the final network configuration may be different for different soil layers.
Reply: 'the' will be removed.

The soil moisture sensing by the current stations provides observations at multiple depths, and are maily centred at 10 cm, 25 cm, 45 cm and 70 cm. The case study presented in the paper is based on the surface soil moisture from the WRF output (at 10 cm). The reviewer is right. The final network configuration could be different for different soil layers. We will, therefore, include the analysis of root zone soil moisture in the updated manuscript.

In Figure 9 it is unclear what is the number of stations considered and the method (CA-max, CA-median) employed to create the "designed network". Please provide more details in the figure caption. Do the current soil moisture monitoring stations only measure soil moisture? Do they observe other hydrological or meteorological variables that need to be taken into account at the time of designing a new network? To account for the network designed in probabilistic terms, have the authors considered using a probabilistic clustering method such as the Fuzzy C-means approach? Does the proposed approach account for the layout of existing monitoring stations?
Reply: The designed network is based on CA-Med and 90% variance contribution rate. This will be added in Figure 9 caption.

The in-situ soil moisture sensors only provide soil moisture information. There are separate rain gauges in the catchment.

We thank the reviewer on suggesting of using probabilistic clustering method such as the Fuzzy C-means approach. This study intends to explore the usefulness of the PCA/CA method which is relatively simple. For future studies, other methodologies will also be explored to examine

their effectiveness. However, we will add the probabilistic clusing method along with other potential approaches as a discussion in the manuscript.

No, the proposed scheme does not account for the layout of existing monitoring stations.

By looking at Figure 10, it seems that a complete re-arrangement of stations is required, which may conflict with existing agency resources and manpower. Judging from Figure 11 it seems that the whole effort of re-designing the network will remove some existing bias, but may not lead to a substantial improvement. Can the authors provide more details on the logic behind the location of existing stations?

Reply: For the current layout of the soil moisture stations, all of the sensors are located in the plain area, which can lead to a bias of the overall catchment soil moisture. However, we can see from Figure 10, some of the existing sensors are located near some of the designed sensors, which could be kept if located within the same cluster. But a lot more sensors are indeed required in the hill zone, where currently no sensors are installed. We will add this discussion in the updated manuscript.

Although in this case study the improvement is not substantial because the soil moisture spatial variations over the catchment are relatively small (see the figure below, from our previous study). However, for areas with large spatial soil moisture variations, the bias could be much noticeable.

[Figure]

Figure 1. The cross-validation of spatially distributed WRF soil moisture against the in-situ soil moisture observation.

The existing stations could be initially installed for irrigation purpose, which hence is mainly located in the plain area.

It seems that the CA-max proposed clustering technique tends to initially place stations at or near the edges of the watershed. While this is the result of the proposed objective method, there are reasons why this might not be ideal. The CA-median seems to be less sensitive to this. How is the PCA superior to defining the number of clusters using something like the Silhouette method? How is the k-means part of this method different than variance minimization methods based on correlograms or semi variograms? How does this approach include or handle the changing land use/land cover?

Reply: CA-Max selects the maximum averaged soil moisture of a cluster. In the case study area, it is clear to see since the southern boundary of the catchment is mainly covered by densely covered trees which generally has higher soil moisture contents than the rest of the catchment, the selected locations tend to distribute near the southern boundary (70% in Figure 6 is a good example). For the CA-Med, as it selects the median averaged soil moisture of a cluster, the resultant locations are more homogeneously distributed. The CA-Max is focused

on the extreme soil moisture condition, whilst the CA-Med is more on the mean condition. Since they provide results in two aspects, it is useful to explore both in the study. The reasoning will be included in the updated manuscript.

Again, we thank the reviewer on suggesting alternative statistical approaches to solve the problem. We will include them in the discussion for future explorations. This paper focuses on assessing the effectiveness of the PCA and K-mean methodologies, and we can see that they result in a very good performance. Moreover, they are simple and can deal with a large number of datasets very efficiently, therefore have the potential to be widely adopted. Since we use the soil moisture temporal variations (10-year daily soil moisture datasets) for the PCA/CA analysis, the local characteristics such as the land use/land cover have already been considered by the analysis and represented by the CA results.

How can this method be applied to smaller catchment areas where a single pixel from remote sensing sources or distributed models is equal or even larger than the entire watershed?
Reply: The method can be applied to smaller catchment areas because the WRF model can simulate high-resolution soil moisture datasets (e.g., 1km, and for small catchments, the computational cost should be low for even higher resolution simulations).